# Transcriptomic and Physiological Studies Unveil that Brassinolide Maintains the Balance of Maize’s Multiple Metabolisms under Low-Temperature Stress

**DOI:** 10.3390/ijms25179396

**Published:** 2024-08-29

**Authors:** Xiaoqiang Zhao, Fuqiang He, Guoxiang Qi, Siqi Sun, Zhenzhen Shi, Yining Niu, Zefeng Wu

**Affiliations:** State Key Laboratory of Aridland Crop Science, College of Agronomy, Gansu Agricultural University, Lanzhou 730070, China; hefq6125@163.com (F.H.); qigx1321@163.com (G.Q.); 15045240973@163.com (S.S.); shizz@gsau.edu.cn (Z.S.); niuyn@gsau.edu.cn (Y.N.); wuzf@gsau.edu.cn (Z.W.)

**Keywords:** maize, low-temperature stress, 2,4-epibrassinolide, RNA-sequencing, WGCNA, high performance liquid chromatography, quantitative real-time PCR

## Abstract

Low-temperature (LT) is one of the major abiotic stresses that restrict the growth and development of maize seedlings. Brassinolides (BRs) have been shown to enhance LT tolerance in several plant species; the physiological and molecular mechanisms by which BRs enhance maize tolerance are still unclear. Here, we characterized changes in the physiology and transcriptome of N192 and Ji853 seedlings at the three-leaf stage with or without 2 μM 2,4-epibrassinolide (EBR) application at 25 and 15 °C environments via high-performance liquid chromatography and RNA-Sequencing. Physiological analyses revealed that EBR increased the antioxidant enzyme activities, enhanced the cell membrane stability, decreased the malondialdehyde formation, and inhibited the reactive oxygen species (ROS) accumulation in maize seedlings under 15 °C stress; meanwhile, EBR also maintained hormone balance by increasing indole-3-acetic acid and gibberellin 3 contents and decreasing the abscisic acid level under stress. Transcriptome analysis revealed 332 differentially expressed genes (DEGs) enriched in ROS homeostasis, plant hormone signal transduction, and the mitogen-activated protein kinase (MAPK) cascade. These DEGs exhibited synergistic and antagonistic interactions, forming a complex LT tolerance network in maize. Additionally, weighted gene co-expression network analysis (WGCNA) revealed that 109 hub genes involved in LT stress regulation pathways were discovered from the four modules with the highest correlation with target traits. In conclusion, our findings provide new insights into the molecular mechanisms of exogenous BRs in enhancing LT tolerance of maize at the seedling stage, thus opening up possibilities for a breeding program of maize tolerance to LT stress.

## 1. Introduction

With the abnormal global climate, cold weather has become more frequent, and the cold spell in spring has lasted longer, which causes a significant disadvantage to seed germination and the seedling growth of crops [1,2]; subsequently, the growth status of seedlings directly affects the yield formation [3]. Maize (*Zea mays* L.), a C_4_ thermophilic crop, tropical in origin, is inherently susceptible to below 15 °C low-temperature (LT) [4]. In early spring, maize seedlings in the northern spring corn area of China are vulnerable to chilling damage of suboptimal temperature of 5–15 °C, in a manner that affects seed germination and possibly results in growth delay, chlorosis, wilting, necrosis, and even death of the seedlings [5,6,7]. At these temperatures, in addition to damage to the photosynthetic organ and decreased light energy utilization, the complex physiological metabolisms can change, including excess reactive oxygen species (ROS) accumulation [8], oxidative stress damage [9], antioxidant defense system alteration [10], hormones imbalance [11], membrane lipid oxidation exacerbation [6], and cold-induced genes activation [2,10,12], in maize seedlings themselves, resulting in yield losses of 20–30% [6,7]. Fortunately, in long-term cold acclimation, maize has evolved its own adaptation mechanism. Therefore, it is necessary to understand the response mechanism of maize to LT stress.

Brassinolides (BRs), a class of efficient, safe, and non-toxic hormones, are a collection of approximately 40 sterol compounds that are ubiquitous in plants [13]. They exhibit distinctive biological effects on normal plant growth and development and resistance to abiotic stresses, including seed germination, cell elongation, dark morphogenesis, flowering, senescence, quality formation, and stress resistance of salt, heavy metals, and drought [13,14,15,16,17,18,19]. Meanwhile, BRs have also attracted extensive attention in alleviating LT stress in recent years. In total, 0.1 mg L^−1^ 2,4-epibrassinolide (EBR) has been proven to reduce the cold damage symptoms of wheat (*Triticum aestivum* L.) by enhancing antioxidant enzyme activities and decreasing malondialdehyde (MDA) formation in the −10 °C environment [20]. Thus, 0.3 mg L^−1^ EBR seed priming prevents rice (*Oryza sativa* L.) seedlings from chilling-induced oxidative stress by maintaining abscisic acid (ABA) and gibberellin 3 (GA_3_) levels at 12/15 °C day/night treatment [21]. Given exposure to 10/5 °C day/night conditions, 0.1 μM EBR improves the photosynthesis of cucumber (*Cucumis sativa* L.) seedlings by promoting the activities of ribulose-1,5-biphosphate carboxylase/oxygenase, fructose-1,6-bisphosphate aldolase, sedoheptulose-1,7-bisphosphatase, and transketolase [22]. However, the effect of BRs on maize seedlings’ growth and physiology, as well as the underlying molecular mechanisms under cold stress, remains unclear.

Empowered by the development of high-throughput RNA-sequencing (RNA-Seq), the molecular mechanisms response to various stress pressures in maize are elucidated, and multiple crucial candidate genes were further identified at a large scale. Additionally, weighted gene co-expression network analysis (WGCNA), a system biology method that characterizes the correlation pattern among genes, facilitates the understanding of gene associations and gene function from RNA-Seq data [23]. So far, this method has been widely employed in abiotic stress resistance analysis in maize, and several reliable hub genes were detected in the studies. For instance, using WGCNA, Liang et al. [24] found four hub genes involved in Ca^2+^ transport in maize in response to salt stress; Zhao et al. [25] determined 19 and 49 hub genes that regulated maize mesocotyl and coleoptile elongation plasticity under different light spectral quality stimulation. Cao et al. [26] identified two co-expression modules with significantly and highly correlated drought-related physiological indicators, with drought-related hub genes including *myeloblastosis* (*MYB*), NAM-ATAF1/2-CUC2 (*NAC*), and basic (region) leucine zippers (*bZIP*) transcription factors (TFs). At present, we know little about gene expression changes affected by EBR-induced maize seedlings under LT stress and corresponding regulating hub genes.

In this study, the physiological study and RNA-Seq analysis were performed in leaves of two maize genotypes seedlings with contrasting tolerance to LT that were treated with or without 2 μM EBR application at 15 and 25 °C temperatures. WGCNA was further used to construct a gene co-expression network. These works try to identify corresponding key differentially expressed genes (DEGs) and hub genes and construct their interconnected networks, which will provide an important theoretical basis for further analyzing the mechanism of exogenous BR in enhancing the LT tolerance of maize seedlings.

## 2. Results

### 2.1. Effects of Exogenous EBR on Physiological Metabolisms under LT Stress

The physiological determination revealed that 15 °C LT stress significantly elevated the ROS levels in leaves of both LT tolerant N192 and LT susceptible Ji853 seedlings at the three-leaf stage. Compared with 25 °C normal temperature (CK), the rate changes (RCs) of O_2_^•−^ production rate, O_2_^•−^ content, and hydrogen peroxide (H_2_O_2_) content in N192/Ji853 leaves under LT stress ranged from 14.5% to 45.4% (Figure 1B). However, the ROS in the leaves of the two maize were significantly decreased when they were treated with 2 μM EBR application at 15 or 25 °C temperatures (LTE or CKE). Comparing leaves of both maize exposed to CK with CKE, their RCs of O_2_^•−^ production rate, O_2_^•−^ content, and H_2_O_2_ content varied from −8.0% to −60.2% (Figure 1A). The same phenomenon was also observed in LTE-treated maize leaves relative to LT-treated leaves; their RCs of ROS varied from −6.6% to −20.3% (Figure 1D), suggesting that EBR application can scavenge ROS accumulation in maize leaves to counteract ROS damage at 15/25 °C environments. Consistently, the MDA content in N192/Ji853 leaves under LT stress was more than twice that in the CK control, which caused a significant decrease of 56.5/73.5% in membrane stability index (MSI) under LT stress (Figure 1B). In contrast, 2 μM EBR induced a 41.1/28.4% decrease in the MDA content of both maize in 15 °C cold imposition, along with a 35.6/26.3% increase in MSI (Figure 1D), indicating that the appropriate EBR application can ameliorate membrane integrity of maize leaves under 15 °C stress. With regard to the antioxidant system, the only exception was catalase (CAT) activity in 15 °C-treated seedlings of Ji853, which decreased by 73.1%, 15 °C stress significantly increased the activities of peroxidase (POD) and glutathione reductase (GR), and the antioxidant content of glutathione (GSH) in their leaves by 18.0~173.2% (Figure 1B). However, only POD and CAT activities in Ji853 leaves increased significantly after EBR application under 15 °C stress, with 40.3% and 417.7%, respectively (Figure 1D). This implies that exogenous EBR can improve the antioxidant capacity of maize seedlings by enhancing the enzymatic and nonenzymatic defense systems under 15 °C conditions. In addition, the most visible difference was that ABA content showed a significant upward trend at 15 °C environment, compared to N192/Ji853 leaves grew in CK environment, with 14.5/47.6% RC; nevertheless, indole-3-acetic acid (IAA) and GA_3_ contents displayed significant downward trend in LT-stressed leaves of N192/Ji853, with −21.5/−29.1% and −14.4/−30.4%, respectively (Figure 1B). Interestingly, 2 μM EBR application under 15 °C stress significantly raised the IAA and GA_3_ contents in N192/Ji853 leaves, relative to 15 °C, with 10.1/16.7% and 9.8/17.8%, respectively; but ABA content reduced slightly, with −2.8/6.9% (Figure 1D), demonstrating that EBR induction can maintain hormone balance by increasing the contents of IAA and GA_3_ and decrease the ABA content in maize seedling under cold stress.

### 2.2. Pearson Correlation and Hierarchical Cluster Analyses

To more effectively identify the correlation between these physiological traits, the Pearson correlation analysis was performed for 12 traits in both maize materials under four treatments. The results showed that there were 62 groups with significant (*p* < 0.05 or *p* < 0.01) positive or negative correlations between both traits (Figure 1E). The findings suggested that the ROS levels, membrane characteristics, antioxidant system, and hormone signaling in maize seedlings interact with each other to enable maize health growth or survival in various environments.

Furthermore, the hierarchical cluster analysis showed that when the Euclidean distance was 7.5, the two maize seedlings under four treatments were clearly divided into four groups (Figure 1F). The Ji853 seedlings growing at 15 °C were the I group, speculating that the LT tolerance of susceptible Ji853 under stress treatment is the worst. N192/Ji853 seedlings treated with 2 μM EBR application at 25 °C were the II group, proving that both maize genotypes grow best when they are exposed to exogenous EBR induction in the control environment. N192 seedlings treated with 2 μM EBR application at 15 °C as well as both genotypes seedlings at 25 °C were the III group, suggesting that the adverse effects of LT stress on tolerant N192 seedlings can be significantly reduced in the treatment of 2 μM EBR application at 15 °C, even reaching normal growth status of both maize genotype at 25 °C environment. N192 seedlings at 15 °C and Ji853 seedlings cultured 2 μM EBR application at 15 °C were the IV group, indicating that the LT tolerance of sensitive Ji853 can be enhanced by EBR stimulation under LT stress, which may be equal to tolerant N192 phenotypes under 15 °C stress.

### 2.3. Evaluation of the RNA-Seq Dataset

To further investigate the transcriptome response to EBR induction in leaves of N192 and Ji853 seedlings at 25 and 15 °C, the BGI DNBSEQ-T7 platform was used to sequence 24 cDNA libraries constructed from high-quality RNA. An average of 49,331,986 clean reads were obtained from each sample, with Q20 and Q30 quality scores > 97% and >93%, respectively, meanwhile the GC content of each sample was approximately 50.76%, and nearly 88% of clean reads mapped to *Zea mays* B73_v4 reference genome (Appendix A).

### 2.4. Defining DEGs

A total of 39,597 unigenes were identified in the 24 databases, which were obviously clustered in nine clusters according to the fragments per kilobase per million mapped (FPKM) expression across two maize genotypes under four treatments (Figure 2A,B). Based on the criteria of |log_2_ fold change (FC)|  >  1, *p*-value < 0.05, and false discovery rate (FDR) < 0.05, 491(N192 leaves in LT_vs_LTE) to 7833 (Ji853 leaves in CK_vs_LT) DEGs were identified among eight comparisons (Figure 2C,D). This suggests that the gene expressions in both tolerant N192 and susceptible Ji853 induced by temperature and exogenous EBR stimulation are different. Surprisingly, only two DEGs (*Zm00001d006768* encoding a C2H2-like zinc finger protein and *Zm00001d018087* encoding an uncharacterized protein) were further identified that were common to six comparisons, as the core conserved DEGs (Figure 2D,E; Appendix A), they may thus be associated with LT tolerance of contrasting maize genotypes under diverse treatments.

### 2.5. Functional Annotation DEGs

To elucidate how EBR induction in maize leaves regulates gene expression to improve LT tolerance under 15 °C stress; the GO annotation and top 20 KEGG enrichments of DEGs in N192 leaves from CK_vs_LT and LT_vs_LTE comparisons were further analyzed. For GO annotation, the main categories were “plasma membrane” on cellular component, “protein serine/threonine kinase activity” and “oxidoreductase activity” on molecular function, and “regulation of defense response”, “response to jasmonic acid (JA) and ABA” on the biological process (Appendix A). This indicates that the DEGs involved in these processes play a significant role in protecting biofilm integrity, plant hormones signal transduction, oxidative stress, and resistance response in N192 leaves. Likewise, the main classification of KEGG was assigned to “plant hormone signal transduction”, “mitogen-activated protein kinase (MAPK) signaling pathway-plant”, “peroxisome”, and “phenylpropanoid biosynthesis” (Appendix A). This suggests that LT stress or EBR application can stimulate multiple pathways in N192 leaves and affect the key gene expression, which then may play important roles in improving the LT tolerance of maize seedlings. Therefore, further studies of the DEGs involved in the above pathways are necessary.

### 2.6. DEGs Involved in ROS Homeostasis

As we know, peroxisomes are a class of simple organelles that play an important role in plant ROS metabolism, which is involved in a series of ROS generation and scavenging [27]. Previous studies had showed that superoxide dismutase (SOD), POD, and CAT are the key antioxidant enzymes for scavenging ROS in plants, and the expression levels of genes encoding these enzymes are significantly correlated with their activities [28]. Therefore, following this, we identified 84 DEGs involved in ROS homeostasis. Of which, 7 were for peroxisome biogenesis, 14 were for fatty acid oxidation, 2 were for ether phospholipid biosynthesis, 1 was for sterol precursor biosynthesis, 11 were for amino acid metabolism, 7 were for the antioxidant system (SOD and CAT) in the peroxisome pathway, and 42 were for CAT in the phenylpropanoid biosynthesis pathway (Figure 3; Appendix A). For example, five positive peroxin (PEX) DEGs involved in peroxisome biogenesis were observed in maize leaves exposed to 15 °C, which may cause the proliferation of peroxisomes. More than 70% of DEGs in fatty acid oxidation showed significant up-regulation in LT-stressed Ji853 leaves relative to the control, with this increase being 1.06-~2.90-fold. This suggests that fatty acid oxidation occurs mainly when Ji853 is subjected to LT stress, subsequently producing massive byproduct ROS. Only two alcohol-forming fatty acyl-CoA reductase (FAR) DEGs in ether phospholipid biosynthesis were identified, i.e., *Zm00001d049950* was up-regulated (6.73-fold) in Ji853 leaves under 15 °C stress, while *Zm00001d019850* was down-regulated (−1.13-fold) in N192 leaves. The diverse expression of FAR genes may thus be necessary for ROS homeostasis in maize under LT stress. *Zm00001d054044* (CAT) showed 1.18- and 1.57-fold up-regulation in N192 leaves from both CK_vs_CKL and CK_vs_LT comparisons; similarly, three SOD DEGs (*Zm00001d036135*, *Zm00001d045538*, and *Zm00001d045384*) showed varied positive expression (1.23-~2.36-fold) in N192 leaves compared to CK_vs_LT; half and nearly 85% DEGs for POD also showed significantly activated expression in Ji853 and N192 leaves from LT_vs_LTE comparison, respectively. This implies that the expression of antioxidant enzyme-related genes increases significantly, thus enhancing the scavenging ability of ROS in maize under LT stress and exogenous EBR mediation.

### 2.7. DEGs Involved in Plant Hormone Signal Transduction

Molecular and physiological responses to temperature stresses are intricately linked to the regulation of hormones [29]. In this study, when the leaves of N192/Ji853 seedlings treated with 2 μM EBR application at 15 and 25 °C, 40 DEGs related to the auxin (AUX) signal pathway, 9 DEGs were associated with the cytokinin (CTK) signal pathway, 14 DEGs were responsible for the GA signal pathway, 34 DEGs were involved in the ABA signal pathway, 10 DEGs were involved in controlling the ethylene (ETH) signal pathway, 8 DEGs participated in the BR signal pathway, 26 DEGs were involved in regulating the JA signal pathway, and 13 DEGs related to the salicylic acid (SA) signal pathway were identified. They showed very complex expression patterns in both maize genotypes leaves under four treatments (Figure 4; Appendix A), suggesting that a variety of hormones act synergistically or antagonistically to regulate maize seedlings’ growth status in various environments, including adaptation to the environment and resistance to stress. In addition, it was noted that 10 up-regulated DEGs involved in all plant hormone signal transduction were found in Ji853 leaves in the LT_vs_LTE comparison, which were more than the identified seven up-regulated DEGs in N192 leaves in the same comparison (Figure 4; Appendix A), suggesting that the exogenous EBR signaling molecule has more significant regulatory effects on genes for hormone signal transduction in sensitive maize genotypes under LT stress to adapt adverse environmental changes.

### 2.8. DEGs Involved in the MAPK Cascade

To coordinate various biotic and abiotic stresses, plants have evolved complex signaling networks to perceive and transmit environmental stimulation, such as MAPK cascade, which is conserved. Universal signal transduction modules play vital roles in controlling intracellular responses to extracellular signals and regulating stress responses in plants [30]. In this study, 89 DEGs associated with MAPK signaling were identified (Figure 5; Appendix A). Of them, a typical MAPK cascade is composed of MAPK kinase kinase (MAPKKK) including four MEKK DEGs, MAPK kinase (MAPKK) including seven MKK DEGs, and MAPK including two MPK DEGs, formed a typical MAPK cascade; they showed significantly varied expression patterns in leaves of both maize genotypes under four treatments. Through stage-by-stage phosphorylation, the MAPK cascade then transmits and amplifies signals to target and regulate multiple resistance genes or TFs (Figure 5). For instance, in Ji853 leaves, *Zm00001d009936* (encoding basic endochitinase B; CHIB) showed −2.08-fold negative expression level in the CK_vs_LT comparison; *Zm00001d018738* (encoding pathogenesis-related protein 1; PR1) showed 1.66-fold positive expression level in the CK_vs_CKE comparison; four *WRKY22* TFs including *Zm00001d011403*, *Zm00001d044171*, *Zm00001d008578*, and *Zm00001d009698* displayed 1.03-~6.05-fold activated expression in the CK_vs_LT comparison. In N192 leaves, *Zm00001d052653* (respiratory burst oxidase; RbohD) had −1.20-fold and −1.33-fold changes in both CK_vs_CKE and CK_vs_LT comparisons; *Zm00001d012482* (*WRKY33* TF) and *Zm00001d034920* (ethylene-responsive transcription factor 1; *ERF1*) had 1.23-fold and 1.76-fold up-regulated expression levels in the LT_vs_LTE comparison, respectively. Moreover, in both N192 and Ji853 leaves, two *MYC* TFs, i.e., *Zm00001d047017* and *Zm00001d030028* were significantly down-regulation in terms of expression levels in the CK_vs_LT comparison (Appendix A).

### 2.9. WGCNA Analysis and Gene Co-Expression Network Construction

To facilitate our understanding of the co-expression network of the physiological metabolisms specific response to LT tolerance in leaves of maize seedlings under LT stress and EBR stimulation, the expression datasets (FPKM > 1) from 24 leaves samples were subjected to WGCNA for finding clusters of gene sets with similar expression patterns (modules). The optimal soft threshold β = 15 (Appendix A) and the network connectivity under different soft thresholds were shown in Appendix A, which thus will be further used to construct the co-expression networks. Subsequently, the similarity matrix was transformed into an adjacency matrix, a topological overlap matrix (TOM), and a dissimilarity matrix in turn, and then the hierarchical clustering generated by the function hierarchical cluster was segmented by dynamic tree cutting. Finally, a total of 14 co-expression modules were obtained (mergeCutHeight = 0.25), of them, the turquoise module contained the most genes, with 1498 genes (Figure 6A,B). Then, the correlations between clusters (modules) using the eigengene module were further performed. Interestingly, the blue module showed the most positive correlation to the O_2_^•−^ production rate (*r* = 0.57), O_2_^•−^ content (*r* = 0.65), H_2_O_2_ content (*r* = 0.66), and MDA content (*r* = 0.85), respectively; the brown module displayed the most positive correlation to MSI (*r* = 0.91), IAA content (*r* = 0.83), and GA_3_ content (*r* = 0.87), respectively; the salmon module had the most positive correlation to POD activity (*r* = 0.86), GR activity (*r* = 0.61), GSH content (*r* = 0.61), and ABA content (*r* = 0.62), respectively; and the red module exhibited the most positive correlation to POD activity (*r* = 0.50) (Figure 6C,D). We, therefore, determine blue brown, salmon, and red modules that are specific modules for corresponding physiological metabolism under LT stress and EBR induction in maize.

Moreover, 66 hub genes in the blue module were identified by setting module membership (MM) > 0.8 and gene significance (GS) > 0.7; 23 hub genes in the brown module were detected by setting MM > 0.8 and GS > 0.8; 12 hub genes in the salmon module were mapped by setting MM > 0.8 and GS > 0.6; and 8 hub genes in the red module were found by setting MM > 0.7 and GS > 0.5 (Appendix A). Furthermore, the GO enrichment analysis showed that the 109 identified hub genes within the four modules were mainly enriched in calcium ion binding, oxidoreductase activity, and stimulus/stress responses (including response to stimulus, cellular response to chemical stimulus, cellular response to heat, response to heat, response to stress, response to abiotic stimulus, response to chemical, and response to temperature stimulus) (Appendix A). Consistently, the KEGG enrichment analysis also revealed that all hub genes were highly significantly enriched in the “MAPK signaling pathway-plant”, “peroxisome”, and “plant hormone signal transduction” (Appendix A). Due to these hub genes within the four modules having different functions, the visualized co-expression network of these hub genes was developed based on a combined score value > 0.8 (Figure 7A). Meanwhile, we also found that the DEGs involved in ROS homeostasis, plant hormone signal transduction, and the MAPK cascade existed in a complex interconnected network (Figure 7B), illustrating that there is a metabolism connection among different pathway communities that represent potential points of crosstalk.

### 2.10. Verification of RNA-Seq Data by Quantitative Real-Time PCR (qRT-PCR)

To validate the accuracy of RNA-Seq data, DEGs encoding SOD (*Zm00001d045538*), POD (*Zm00001d046184* and *Zm00001d047358*), CAT (*Zm00001d054044*), abscisic acid receptor PYL9 (PYR/PYL; *Zm00001d016105*), serine/threonine-protein kinase SRK2A (SnRK2; *Zm00001d026690*), protein phosphatase 2C 16 (PP2C; *Zm00001d038846*), WRKY 33 (*Zm00001d012482*), and calmodulin-7 (*Zm00001d028948*) were randomly selected for qRT-PCR analysis. The results showed that the qRT-PCR expression patterns were in agreement with the relevant nine DEGs in the RNA-Seq dataset, and there was a good linear relationship between the qRT-PCR expression levels and RNA-Seq dataset in leaves of both N192 and Ji853 seedlings under four treatments (y = −0.593 + 0.663x; *R*^2^ = 0.514**) (Figure 8), reflecting the reliability of the RNA-Seq data.

## 3. Discussion

Maize is particularly susceptible to LT, and improving its LT tolerance is of great significance to stabilize yield and enhance food security by broadening the geographical regions in which maize can be cultivated. LT stress can destroy cellular homeostasis, cause oxidative stress, and cause the accumulation of high levels of ROS in maize cells [31]. Similar results were obtained in our study in terms of exposure to 15 °C stress, as evidenced by increased O_2_^•−^ production rate (33.0%), O_2_^•−^ content (45.4%), and H_2_O_2_ content (16.6%) in N192 and Ji853 leaves. It is known that MDA is a crucial indicator of damage caused by membrane lipid peroxidation and the ability of cells to withstand LT stress [32]. Under stress of 15 °C, increased MDA content (161.8%) and decreased MSI (65.0%) indicated that the membrane lipids of maize were subjected to severe cold damage in this study. Previous reports showed that maize leaves subjected to 4–15 °C stress severe lipid damage can present with a pale green chlorotic appearance, premature leaf senescence, and even seedling death [6,33], and Magnolia’s research also supported this view [34]. Therefore, this possibly explains the increased maize seedlings mortality observed at below 15 °C environment. Fortunately, a clear suppression of oxidative stress damage by maintaining higher activities of antioxidant enzymes in wheat during cold acclimation [35] was further confirmed by our data: 15 °C stress significantly drove antioxidant systems, as exemplified by the increased POD (1.7 times) and GR (2.1 times) activities, as well as GSH content (1.4 times) in stressed-leaves of both maize genotypes. However, the most distinct discordance was also noted for the obvious decrease in CAT activity in Ji853 leaves at a 15 °C environment, with 73.1%, which may be related to the damage of the antioxidant system of cold-sensitive maize under long-term LT stress. In addition, hormones play an important role in coordinating the growth of plants and their tolerance to LT. Our study showed that 15 °C stress increased the ABA level (31.0%) and decreased IAA (25.3%) and GA_3_ (22.4%) levels in both maize leaves. Similarly, the hormone content assays showed an increase in ABA content in young wheat ears but a decrease in IAA and GA contents under 4 °C stress for 60 h with regard to wheat grain number at the booting stage [36]. We thus speculated that LT stress may affect the biosynthesis, decomposition, transport, and stability of hormones in cereal crops.

With a few exceptions, BRs have been shown to improve plant adaptations to biotic and abiotic stresses [37]; however, the mechanisms of BR action in enhancing maize tolerance to LT stresses still remains largely unknown. As a first line of defense, the antioxidant system, i.e., antioxidant enzymes activities of SOD, POD, and CAT activities, increased by 18.3%, 10.7%, and 21.4%, and the antioxidant contents of GSH and oxidized glutathione (GSSG) increased by 16.1% and 44.7% in BR-mediated seedling in rice under 10 °C stress, while concomitantly reducing H_2_O_2_ content with 11.3% [38]. Likewise, 0.1 mg L^−1^ EBR treatment significantly decreased the O_2_^•−^ and H_2_O_2_ contents in grapevine seedlings in a 4 °C environment; this is primarily related to the improved antioxidant system, the obvious increase in SOD, GR, dehydroascorbic acid reductase (DHAR), and monodehydroascorbate reductase (MDHAR) activities were observed after EBR application [39]. Consistently, our study showed that 2 μM EBR stimulation significantly increased POD and CAT activities by 10.1% and 199.2% in both maize leaves at 15 °C environment, whereas their ROS production, including O_2_^•−^ (16.2%) and H_2_O_2_ (10.0%) contents in leaves, were inhibited significantly. Thus, BR exerts a protective effect by improving the antioxidant system against adverse reactions resulting from cold stress [40]. Meanwhile, our results showed that 2 μM EBR application clearly inhibited MDA (34.7%) accumulation in maize leaves under 15 °C stress, resulting in a 1.3 times increase by MSI. Li et al. [41] also revealed that plasma membrane lipids in 10 μM BR-treated fruit showed lower phase transition temperature and higher unsaturation degree, leading to higher fluidity under 4 °C stress. Therefore, after LT stress, BR-mediated maize leaves can inhibit membrane lipid peroxidation and safeguard cellular membrane integrity. Notably, an early study reported that soaking with 0.1 mg L^−1^ BR maintained hormone balance by increasing the contents of IAA, SA, and zeatin (ZT) and decreasing ABA content in rice seedlings under NaCl stress [42]. A similar study also found that after 0.1 mg L^−1^ BR application under weak light stress, IAA and GA contents of tomato (*Lycopersicon esculentum* L.) leaves increased significantly; instead, this occurred along with a clear decrease in ABA level [43]. The same with these studies, our results also confirmed that 2 μM EBR imposition in maize leaves significantly increased IAA and GA_3_ contents by 13.4% and 13.8% under cold stress of 15 °C, whereas they slightly decreased ABA content by 4.8%. So,, these findings suggested that BR stimulation can regulate hormone formation and be involved in the hormone balance of maize seedlings to improve their LT tolerance and maintain normal functions.

Here, we performed an RNA-Seq to further explore the molecular mechanisms of EBR mediation enhancing LT tolerance of maize seedlings. For the leaves of both N192 and Ji853 seedlings, the CK_vs_LT comparison yielded the largest number of DEGs, followed by the CK_vs_LTE and LT_vs_LTE comparisons. This was well supported by the Venn analysis of DEGs by Luo et al. [44], demonstrating that when maize seedlings were subjected to 15 °C stress, 2 μM EBR application can increase DEG numbers. This study further revealed the enrichment terms of these DEGs through GO and KEGG analyses, such as “oxidoreductase activity”, “protein serine/threonine kinase activity”, “plant hormone signal transduction”, “MAPK signaling pathway-plant”, “peroxisome”, and “phenylpropanoid biosynthesis”. Similarly, in foxtail millet (*Setaria italica* L.) [45] and tea plants (*Camellia sinensis* L.) [46] subjected to LT stress with suitable EBR application, the KEGG enrichment analysis suggested the DEGs associated with “plant hormone signal transduction” and “MAPK signaling pathway-plant” pathways. These findings indicated that exogenous BR can induce differential expression of multiple genes in the above metabolic pathways to alleviate symptoms of chilling injury in maize.

Peroxisomes are highly dynamic and metabolically active organelles that mainly participate in ROS metabolism. In the peroxisome pathway, the *Zm00001d012262* (PEX3) was significantly down-regulated in EBR-treated N192 leaves at 15 °C. Rodríguez-Serrano et al. [47] reported that *Arabidopsis thaliana PEX11a* RNAi lines were unable to produce peroxules under stress conditions. Lismont et al. [48] found that the inactivation of *PEX11B* led to partial localization of both peroxisomal membrane and matrix proteins to mitochondria and a decrease in peroxisome density. These findings demonstrated that *PEX* genes may play a key role in regulating stress perception and fast cell responses to environmental cues. Until now, it has been widely assumed that H_2_O_2_ could freely permeate peroxisomal membrane through peroxisomal membrane protein 2 (PXMP2; a non-selective pore-forming protein with an upper molecular size limit of 300–600 Da) [48]. In this study, only one PXMP2 DEG, *Zm00001d016305,* had a significantly negative expression level of Ji853 leaves in the CK_vs_LT comparison. It is assumed that LT stress does not facilitate H_2_O_2_ penetration in the peroxisome membrane. Katayose et al. [49] generated single and double mutants of long-chain acyl-CoA synthetase (ACSL) genes in *Marchentia polymorpha* L., i.e., *MpACSL1* and *MpACSL2*, and revealed that they were involved in stress responses. Interestingly, the *Zm00001d023351* for ACSL showed varied expression patterns in maize leaves under both LT stress and EBR induction. This informs us that the expression level of this ACSL gene in maize is induced by EBR stimulation and LT stress. Isocitrate dehydrogenase (IDH) is a key enzyme controlling the activity of the citric acid (TCA) cycle [50], and under drought stress, 50 mM citric acid application to tobacco (*Nicotiana tabacum* L.) plants maintained a better ROS homeostasis through increased activity of antioxidant enzymes and decreased H_2_O_2_ content [51]. In this study, one activated IDH DEG, *Zm00001d021772,* was identified in maize leaves under stress and EBR treatments, which may rely on the TCA cycle to positively regulate ROS homeostasis in LT-stressed/EBR-treated maize. Moreover, a series of antioxidant-coded genes, such as *Fe/Mn-SOD*, *CAT2*, *APX1*, *APX7*, *GSTU6*, and *GST4,* in wheat were up-regulated under drought stress with 10 μM strigolactones (SLs) priming [52]. Similarly, three SOD (*Zm00001d025106*, *Zm00001d014632*, and *Zm00001d045384*), three POD (*Zm00001d013212*, *Zm00001d028347*, and *Zm00001d045845*), and one CAT (*Zm00001d021772*) DEGs were up-regulated in Ji853 leaves under 15 °C stress with 2 μM EBR application, to effectively scavenge ROS.

The hormone signal transduction pathway plays an important role in a plant’s resistance to stress environments [53]. In this study, the expression of *Zm00001d009309* (xyloglucosyl transferase TCH4, TCH4), a BR biosynthetic gene, was up-regulated in maize leaves under LT stress and 2 μM EBR induction stabilized its expression level; this might be explained by the down-regulation of *Zm00001d043634* (brassinosteroid insensitive 1, BRI1, involving extracellular BR signal perception) under LT stress. Similarly, the *Arabidopsis thaliana bri1* mutation resulted in defective BR signaling and generated an increased tolerance to cold [54]. Consistent with the results of exogenous glutathione enhancing tolerance of the potato (*Solanum tuberosum* L.) to cadmium stress [53], our study revealed that 2 μM EBR application under 15 °C stress in maize leaves down-regulated the expression of the auxin influx carrier gene (AUX1; *Zm00001d038275*), activated transcription inhibitor auxin-responsive protein IAA (AUX/IAA; *Zm00001d052934*, *Zm00001d010360*, *Zm00001d039513*, and *Zm00001d033976*), and inhibited auxin response factor (ARF; *Zm00001d003601*), thereby inhibiting AUX signal transduction and enhancing LT tolerance of maize. Notably, the exogenous addition of CTK may increase freezing tolerance in rice and stabilize cell membranes [55]. Except for 10 mg L^−1^ diethyl aminoethyl hexanoate (DA-6) pretreatment up-regulating CTK biosynthesis-related genes and down-regulating its inactivation genes in tomato plants under low night temperature [56], our transcriptome in maize leaves also indicated that CTK signal-related genes, such as *Zm00001d045112* (two-component response regulator ARR-B family, B-ARR) and *Zm00001d003598* (two-component response regulator ARR-A family, A-ARR) show significant inhibition expression; thus, they may be involved in improving LT tolerance in maize with 2 μM EBR application under LT stress. GID1 (a GA receptor) interacts with the DELLA protein (a repressor of GA signaling) in a GA-dependent manner [57]. Our study revealed that 2 μM EBR application increased GID1 (*Zm00001d038165*) gene expression and inhibited DELLA (*Zm00001d013465* and *Zm00001d033680*) gene expression in N192 leaves during cold stress of 15 °C, which increased GA_3_ accumulation and enhanced LT tolerance. PYR/PYL, SnRK2, and ABA-responsive element binding factor (ABF) are important components of the ABA signal pathway [58]. In this study, we showed that 2 μM EBR stimulation at 15 °C inhibited PP2C (*Zm00001d012401*, *Zm00001d005609*, and *Zm00001d020100*) genes expression by up-regulating the gene expression of its receptor PYR/PYL (*Zm00001d053396* and *Zm00001d016105*), which eventually increased SnRK2 (*Zm00001d011604*) and ABF (*Zm00001d022550* and *Zm00001d012273*) gene expression. This suggests that PYR/PYL is a positive regulator underlying the alleviation of LT stress in maize by EBR induction. Like potatoes that were treated with 0.5 μM EBR pretreatment under drought stress [59], our study showed that 2 μM EBR increased the gene expression of ethylene receptor (ETR; *Zm00001d051889*) and ethylene-responsive transcription factor 1 (ERF1/2; *Zm00001d034920*) in N192 leaves under 15 °C stress. This leads us to speculate that these genes may positively regulate the response to diverse stress following EBR treatment. The jasmonate ZIM domain-containing protein (JAZ), as a negative regulator, affects the transduction of the JA signal by interacting with the MYC2 TFs [60]. Early studies reported that the JA pathway inhibited the transcriptional activity of ICE1/2 via the JAZ protein and regulated the LT tolerance of *Arabidopsis thaliana* [61]. In our study, 2 μM EBR application enhanced JA signal transduction by inhibiting JAZ (*Zm00001d005726* and *Zm00001d048268*) gene expression in maize under cold stress. In addition, Ji853 leaves were exposed to 2 μM EBR treatment under 15 °C stress; we also found that related TGA TFs (*Zm00001d052543* and *Zm00001d005143*) were up-regulated, thus enhancing SA signal transduction.

Increasing evidence has indicated that genetic manipulation of the abundance or activity of some MAPK components can enhance tolerance to many stresses in crops [62]. Overall, 90% of the examined MAPK cascade genes were induced under cold stress and 60% of the genes were induced by high-temperature stress in *Brachypodium distachyon* L. [63]. So far, there have been 24 MAPK, 10 MAPKK, and 72 MAPKKK components identified in the maize genome [64], while 13 DEGs for 2 MAPK, 7 MAPKK, and 4 MAPKK were found in both maize genotypes under 15 °C stress and 2 μM EBR stimulation from our study, which only account for 12.3%. This suggests that partial MAPK cascade genes can be induced by LT stress and EBR stimulation in maize. At the same time, among the TFs, numerous WRKY TFs have been evaluated in plants for their crucial role in the regulation of stress responses. Guo et al. [65] reported that *GhWRKY59* regulated MAPK activation and *GhMAPKKK* expression through feedback in cotton (*Gossypium hirsutum*). Wang et al. [66] found that the *GhWRKY40* transcript level was increased by methyl jasmonate (MeJA), SA, and ET induction; meanwhile, it also regulated *GhMAP3K15* expression positively by establishing a feedback loop. In this study, 8 WRKY TFs in the MAPK cascade showed significant up-/down-regulation in maize leaves by the 15 °C stress and 2 μM EBR induction. These findings thus highlight the importance of WRKYs interacting with the MAPK cascade in regulating diverse stress and hormone responses.

Using WGCNA, four specific modules (blue, brown, salmon, and red) related to LT tolerance of N192/Ji853 leaves were identified in this study. By calculating the MM and GS values of the eigengenes in these modules, a total of 109 hub genes were finally identified. It is speculated that they may play important roles in the LT tolerance of maize at the seedling stage. Further analysis showed that 88 of 109 hub genes and 225 DEGs involved in several key pathways (such as ROS homeostasis, plant hormone signal transduction, Ca signal transduction, and MAPK cascade) formed the complex interaction networks. Therefore, we speculated that maize has complex stress-regulation mechanisms to protect maize seedlings from damage under LT stress.

On the basis of our findings and those reported in previous studies, we have tried to provide insights into the physiological and molecular mechanisms underlying the alleviation of LT stress by 2 μM EBR stimulation in maize seedlings (Figure 9). Specifically, EBR application induced genes/TFs expression of ROS homeostasis, plant hormone signal transduction, Ca signal transduction, and MAPK cascade, which meanwhile interacted with each other to maintain multiple metabolisms balances, resulting in the O_2_^•−^ production rate, O_2_^•−^ content, H_2_O_2_ content, MDA accumulation, and ABA level decreasing, while POD activity, CAT activity, the IAA level, and the GA_3_ level were increased in maize leaves with 2 μM EBR application under LT stress. Collectively, the physiological and molecular changes led to a beneficial response to LT stress in EBR-treated seedlings.

## 4. Materials and Methods

### 4.1. Experimental Materials and Treatments

LT tolerant N192 and LT susceptible Ji853 maize genotypes were used in this study. Their seeds were disinfected in 0.5% (*v*/*v*) sodium hypochlorite solution for 10 min and washed three times with double-distilled water (ddH_2_O) to remove the residue of disinfectant; they were then primed with ddH_2_O at 20 °C in darkness for 24 h. Subsequently, ten seeds were sown into pots (height 17 cm, diameter 20 cm) filled with sterilized vermiculite, which were grown in a growth chamber with a 12 h light/12 h dark photoperiod and constant temperature (25 °C) and humidity (70%). When the seedlings grew three leaves, they were placed at 25 and 15 °C for 7 days and sprayed with 5 mL 0 (ddH_2_O) or 2 μM EBR (CAS: 78821-43-9; Sigma-Aldrich Ltd., Shanghai, China) solution each day. Collectively, four treatments, namely CK treatment (0 μM EBR application at 25 °C normal temperature), CKE treatment (2 μM EBR application at 25 °C normal temperature), LT treatment (0 μM EBR application at 15 °C LT stress), and LTE treatment (2 μM EBR application at 15 °C LT stress), with three replicates were conducted. Then, the 3rd leaf of both genotypes seedlings under all treatments was collected and stored at −70 °C for physiological metabolism measurements, RNA-Seq analysis, and qRT-PCR gene verification.

### 4.2. Physiological Metabolism Measurements

The O_2_^•−^ production rate, O_2_^•−^ content, H_2_O_2_ content, MDA content, GSH content, POD activity, CAT activity, and GR activity were determined using corresponding Solarbio kits (Beijing Solarbio Science and Technology Co., Ltd., Beijing, China; http://www.solarbio.com, accessed on 6 January 2024) and using the multi-function microplate reader (SynergyHTX; BioTek Instruments, Inc., Winooski, VT, USA), following the manufacturer’s instructions, respectively. Refer to the method of Zhao et al. [6]. The 0.2 g fresh leaves were placed into the test tubes containing 10 mL ddH_2_O and incubated at 40 °C for 30 min. The conductivity (C1) was measured using a DDSJ-308F conductivity meter (Rex Electric Chemical, Shanghai, China). Then, the same set was kept in a water bath at 100 °C for 15 min, and the conductivity (C2) was recorded. The MSI was estimated as follows [6]:MSI = [1 − (C1/C2)] × 100%(1)

Refer to the method of Zhao et al. [25] via high-performance liquid chromatography (HPLC). The 0.5 frozen leaves were ground in liquid nitrogen and digested in 5 mL methanol-formic acid solution (99/1, *v*/*v*) at 4 °C for 12 h, then centrifuged at 12,000 rpm (Centrifuge 5425/5425 R; Eppendorf, Hamburg, Germany) at 4 °C for 20 min, and the supernatant was collected. The IAA content, ABA content, and GA_3_ content were determined using Aglient 1100, 1260 (Palo Alto, CA, USA).

### 4.3. Statistical Analyses

For all physiological metabolisms of N192/Ji853 leaves under four treatments, their rate changes (RC) from different comparisons (CK_vs_CKE, CK_vs_LT, CK_vs_LTE, and LT_vs_LTE) were calculated as follows [6]:RC_(CK_vs_CKE)_ = (T_CKE-i_ − T_CK-i_)/T_CK-i_ × 100%(2)
RC_(CK_vs_LT)_ = (T_LT-i_ − T_CK-i_)/T_CK−i_ × 100%(3)
RC_(CK_vs_LTE)_ = (T_LTE-i_ − T_CK-i_)/T_CK-i_ × 100%(4)
RC_(LT_vs_LTE)_ = (T_LTE-i_ − T_LT-i_)/T_LT-i_ × 100%(5)
where T_CK-i_, T_CKE-i_, T_LT-i_, and T_LTE-i_ were the *i*-th physiological metabolism of a single maize genotype under CK, CKE, LT, and LTE treatment, respectively. Pearson correlation analysis was performed using the online Genescloud tool (https://www.genescloud.cn, accessed on 12 April 2024). Hierarchical cluster analysis with between-group linkage was performed using the R package v.1.17-4 (http://www.R-project.org, accessed on 13 April 2024).

### 4.4. RNA-Seq and DEG Identification

The leaves of CK, CKE, LT, and LTE treatments in both maize genotypes were collected with three replicates, and RNA-Seq was performed using the BGI DNBSEQ-T7 system by Shanghai Personal Biotechnology Co., Ltd., Shanghai, China. After sequencing, the clean reads were obtained and aligned to the *Zea mays* B73_v4 reference genome (ftp://ftp.ensemblgenomes.org/pub/plants/release-6/fasta/zea_mays/dna/, accessed on 29 August 2023) using HISAT 2.2.1 (http://ccb.jhu.edu/software/hisat2, accessed on 29 August 2023). The data were analyzed using HTSeq v.0.9.0 (http://htseq.readthedocs.io/en/release_0.9.1/, accessed on 29 August 2023) based on the readcount data obtained from expression profiling and calculated the fragments per kilobase per million mapped (FPKM) [67]. The DEGs were defined using the criteria |log_2_ FC|  >  1, *p*-value < 0.05, and FDR < 0.05 and identified using the DESeq2 package (http://bioconductor.org/packages/release/bioc/html/DESeq.html, accessed on 29 August 2023). For identified DEGs, GO enrichment analysis was performed using AmiGO 2 database (http://amigo.geneontology.org/amigo/, accessed on 29 August 2023); KEGG analysis was performed using the KEGG database (https://www.kegg.jp/kegg/, accessed on 29 August 2023).

### 4.5. WGCNA Analysis and Network Construction

After filtering the required genes using the function of goodSamplesGenes provided by the R package WGCNA v.1.72-5 (Los Angeles, CA, USA; https://cran.r-project.org/web/packages/WGCNA/index.html, accessed on 16 March 2024), WGCNA [68] was performed to obtain a more accurate co-expression network. The soft-threshold power was determined based on the principle of scale-free networks, and the software-provided soft-threshold power was used for subsequent analysis. The mergeCutHeight = 0.25 was used to merge similar modules. A module was defined as significant if the *p*-value for module-trait association was 0.05 [25]. Eigenvector gene analysis was used to identify target gene modules and cluster analysis of eigenvector genes (module eigengenes, ME) was performed in all modules [69]. The ME correlation between modules was also performed to further confirm the correlation between modules and ME. According to the MM and GS values to screen hub genes in specific modules. The co-expression network of all hub genes and interaction network of DEGs in key pathways were constructed using Cytoscape v.3.7.1 (https://cytoscape.org/, accessed on 10 June 2024).

### 4.6. qRT-PCR Validation

Nine random DEGs were selected for validation by qRT-PCR. Single-stranded cDNA was obtained using the PrimeScript^TM^ 1st strand cDNA synthesis Kit (TaKaRa, Japan). Gene special primers were designed using the Primer3web v.4.1.0 (https://primer3.ut.ee/, accessed on 16 June 2024) (Appendix A). The LightCycler480II fluorescent quantitative PCR instrument (Roche, Munich, Germany) was used for qRT-PCR analysis. The *ZmACT-1* was used as the internal control for normalization [25]. There were three replicates for gene relative expression analysis, and the relative gene expression level was estimated by the 2^−ΔΔCT^ method [25]. The analysis of variance (ANOVA) of relative expression levels was performed using IBM-SPSS Statistics v.19.0.

## 5. Conclusions

In conclusion, EBR application effectively enhanced LT tolerance in maize seedlings. EBR induced significant expression of genes/TFs involved in ROS homeostasis, plant hormone signal transduction, Ca signal transduction, and MAPK cascade in maize seedlings under LT stress. These molecular changes maintained multiple physiological metabolisms including oxidative stress damage, ROS level, antioxidant defense system, hormone balance, and membrane lipid peroxidation and stability for protection against cold stress. These findings enhance our contribution to the understanding of the mechanism by which exogenous EBR enhances the LT tolerance of maize and the cold-responsive genes identified following the application of EBR provide targets for future molecular breeding.

## Figures and Tables

**Figure 1 ijms-25-09396-f001:**
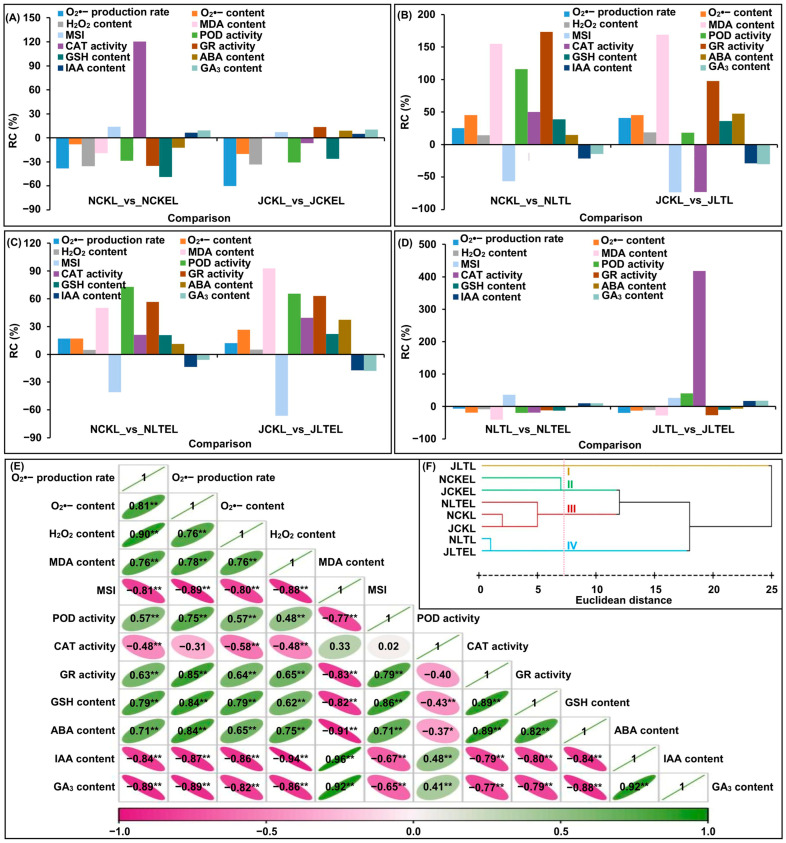
The rate changes (RCs) and relationships of twelve physiological traits in leaves of both N192 and JI853 seedlings at the three-leaf stage under four treatments. MDA: malondialdehyde; MSI: membrane stability index; POD: peroxidase; CAT: catalase; GR: glutathione reductase; GSH: glutathione; ABA: abscisic acid; IAA: indole-3-acetic acid; GA_3_: gibberellin 3. NCKL: leaves of N192 seedlings treated with 0 μM 2,4-epibrassinolide (EBR) application at 25 °C normal temperature (CK); NCKEL: leaves of N192 seedlings treated with 2 μM EBR application at 25 °C normal temperature (CKE); NLTL: leaves of N192 seedlings treated with 0 μM EBR application at 15 °C low-temperature (LT); NLTEL: leaves of N192 seedlings treated with 2 μM EBR application at 15 °C normal temperature (LTE); JCKL: leaves of Ji853 seedlings at CK; JCKEL: leaves of Ji853 seedlings at CKE; JLTL: leaves of Ji853 seedlings at LT; JLTEL: leaves of Ji853 seedlings at LTE. (**A**) RCs of all traits in NCKL_vs_NCKEL and JCKL_vs_JCKEL comparisons. (**B**) RCs of all traits in NCKL_vs_NLTL and JCKL_vs_JLTL comparisons. (**C**) RCs of all traits in NCKL_vs_NLTEL and JCKL_vs_JLTEL comparisons. (**D**) RCs of all traits in NLTL_vs_NLTEL and JLTL_vs_JLTEL comparisons. (**E**) Pearson correlation coefficient analysis. */** represented the *p* < 0.05/*p* < 0.01 level of significant correlation between both traits, respectively. (**F**) Hierarchical cluster analysis.

**Figure 2 ijms-25-09396-f002:**
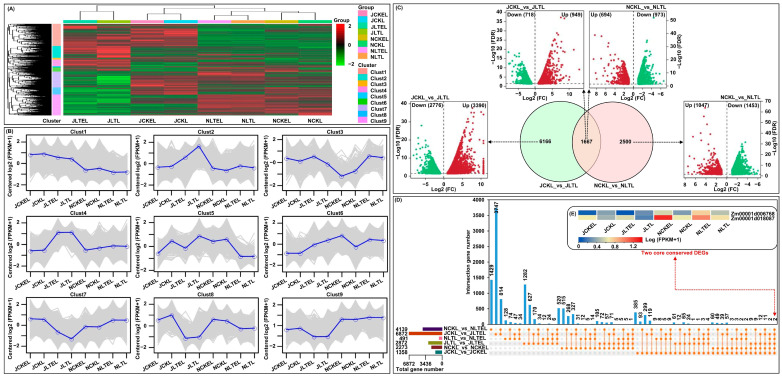
Expression profiles of all unigenes and differentially expressed genes (DEGs) were identified in leaves of both N192 and JI853 seedlings at the three-leaf stage under four treatments. NCKL: leaves of N192 seedlings treated with 0 μM 2,4-epibrassinolide (EBR) application at 25 °C normal temperature (CK); NCKEL: leaves of N192 seedlings treated with 2 μM EBR application at 25 °C normal temperature (CKE); NLTL: leaves of N192 seedlings treated with 0 μM EBR application at 15 °C low-temperature (LT); NLTEL: leaves of N192 seedlings treated with 2 μM EBR application at 15 °C normal temperature (LTE); JCKL: leaves of Ji853 seedlings at CK; JCKEL: leaves of Ji853 seedlings at CKE; JLTL: leaves of Ji853 seedlings at LT; JLTEL: leaves of Ji853 seedlings at LTE. (**A**) Heat map showing the expression profiles of all unigenes to obtain nine gene clusters. (**B**) The representative expression patterns of nine gene clusters. (**C**) Venn diagram and volcano plots showing DEGs in two comparisons. FC: fold change; FDR: false discovery rate. (**D**) Upsetplot showing the DEGs among six comparisons to screen out two core conserved DEGs. (**E**) Heat map showing the expression patterns of the two core conserved DEGs in all samples. FPKM: the fragments per kilobase per million mapped.

**Figure 3 ijms-25-09396-f003:**
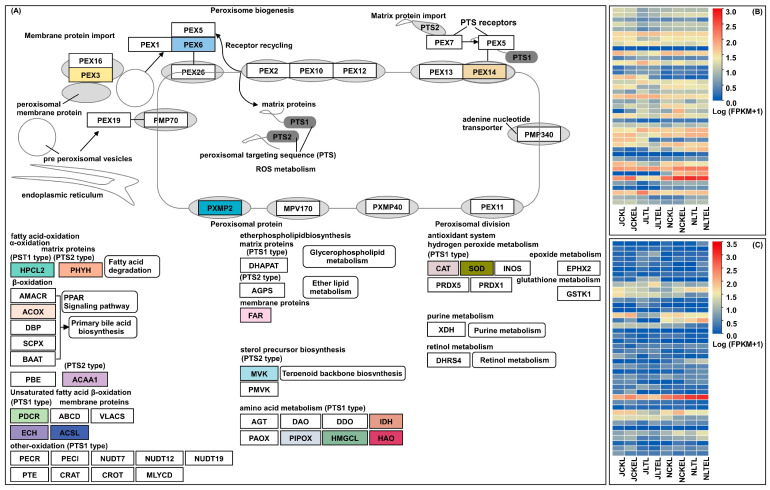
The peroxisome pathway and expression profiles of differentially expressed genes (DEGS) involved in reactive oxygen species (ROS) homeostasis in leaves of both N192 and JI853 seedlings at the three-leaf stage under four treatments. NCKL: leaves of N192 seedlings treated with 0 μM 2,4-epibrassinolide (EBR) application at 25 °C normal temperature (CK); NCKEL: leaves of N192 seedlings treated with 2 μM EBR application at 25 °C normal temperature (CKE); NLTL: leaves of N192 seedlings treated with 0 μM EBR application at 15 °C low-temperature (LT); NLTEL: leaves of N192 seedlings treated with 2 μM EBR application at 15 °C normal temperature (LTE); JCKL: leaves of Ji853 seedlings at CK; JCKEL: leaves of Ji853 seedlings at CKE; JLTL: leaves of Ji853 seedlings at LT; JLTEL: leaves of Ji853 seedlings at LTE. FPKM: the fragments per kilobase per million mapped. (**A**) Peroxisome pathway. (**B**) Expression profiles of DEGs in peroxisome pathway. (**C**) Expression profiles of DEGs for peroxidase in phenylpropanoid biosynthesis pathway.

**Figure 4 ijms-25-09396-f004:**
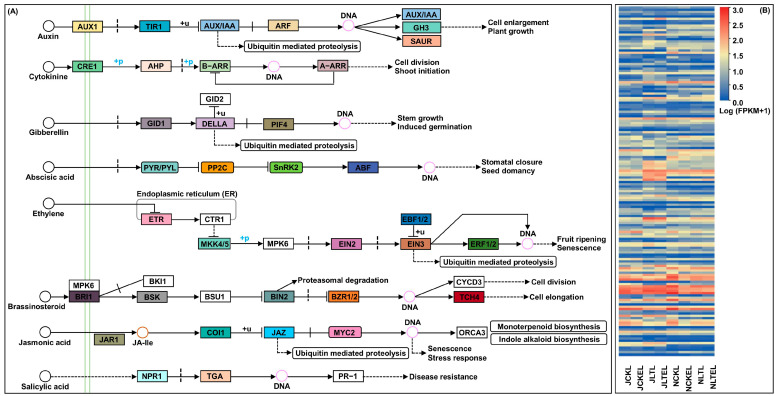
The plant hormone signal transduction pathways and expression profiles of differentially expressed genes (DEGS) involved in eight plant hormones signal transduction in leaves of both N192 and JI853 seedlings at the three-leaf stage under four treatments. NCKL: leaves of N192 seedlings treated with 0 μM 2,4-epibrassinolide (EBR) application at 25 °C normal temperature (CK); NCKEL: leaves of N192 seedlings treated with 2 μM EBR application at 25 °C normal temperature (CKE); NLTL: leaves of N192 seedlings treated with 0 μM EBR application at 15 °C low-temperature (LT); NLTEL: leaves of N192 seedlings treated with 2 μM EBR application at 15 °C normal temperature (LTE); JCKL: leaves of Ji853 seedlings at CK; JCKEL: leaves of Ji853 seedlings at CKE; JLTL: leaves of Ji853 seedlings at LT; JLTEL: leaves of Ji853 seedlings at LTE. FPKM: the fragments per kilobase per million mapped. (**A**) Plant hormone signal transduction pathway. (**B**) Expression profiles of DEGs in eight plant hormone signal transduction pathways.

**Figure 5 ijms-25-09396-f005:**
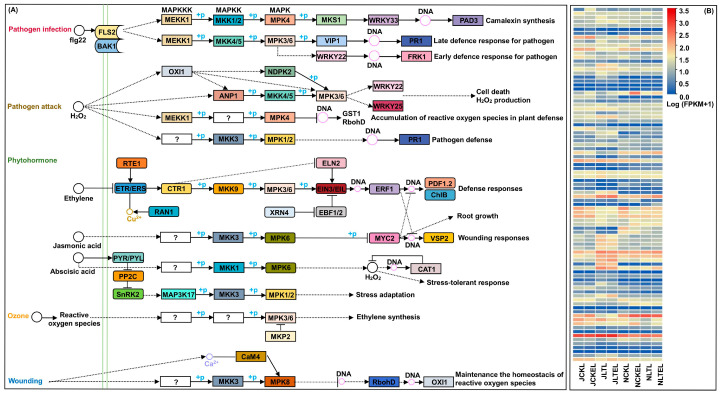
The MAPK signaling pathway and expression profiles of differentially expressed genes (DEGS) involved in the MAPK signaling pathway in leaves of both N192 and JI853 seedlings at the three-leaf stage under four treatments. NCKL: leaves of N192 seedlings treated with 0 μM 2,4-epibrassinolide (EBR) application at 25 °C normal temperature (CK); NCKEL: leaves of N192 seedlings treated with 2 μM EBR application at 25 °C normal temperature (CKE); NLTL: leaves of N192 seedlings treated with 0 μM EBR application at 15 °C low-temperature (LT); NLTEL: leaves of N192 seedlings treated with 2 μM EBR application at 15 °C normal temperature (LTE); JCKL: leaves of Ji853 seedlings at CK; JCKEL: leaves of Ji853 seedlings at CKE; JLTL: leaves of Ji853 seedlings at LT; JLTEL: leaves of Ji853 seedlings at LTE. FPKM: the fragments per kilobase per million mapped. (**A**) MAPK signaling pathway. (**B**) Expression profiles of DEGs in MAPK signaling pathway.

**Figure 6 ijms-25-09396-f006:**
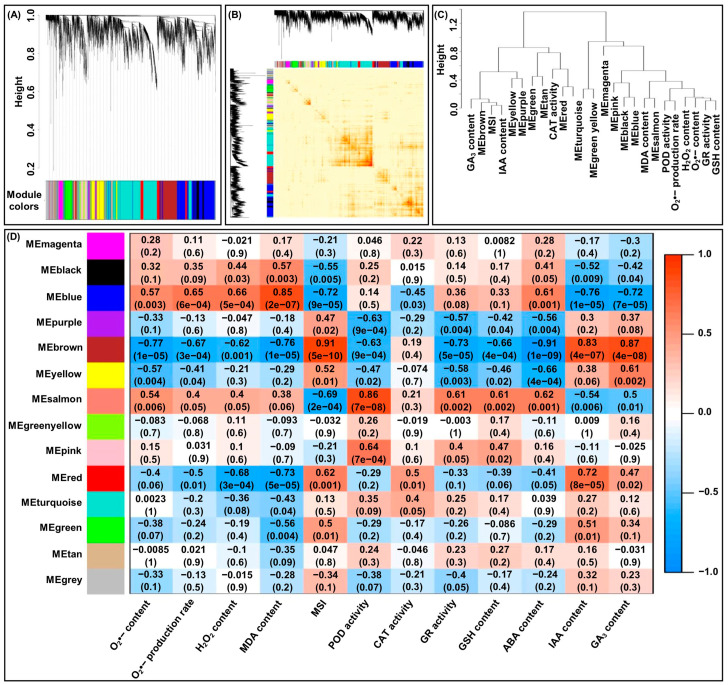
The WGCNA co-expression analysis and module-trait correlation analysis. (**A**) Hierarchical cluster tree showing co-expression modules by the dynamic tree cutting. Each line represents one gene. The module color underneath the cluster tree shows the result of the module assignment. (**B**) Correlated heat maps between modules. (**C**) Module eigengene (ME) cluster tree. (**D**) Correlation of physiological metabolisms with WGCNA modules. MDA: malondialdehyde; MSI: membrane stability index; POD: peroxidase; CAT: catalase; GR: glutathione reductase; GSH: glutathione; ABA: abscisic acid; IAA: indole-3-acetic acid; GA_3_: gibberellin 3.

**Figure 7 ijms-25-09396-f007:**
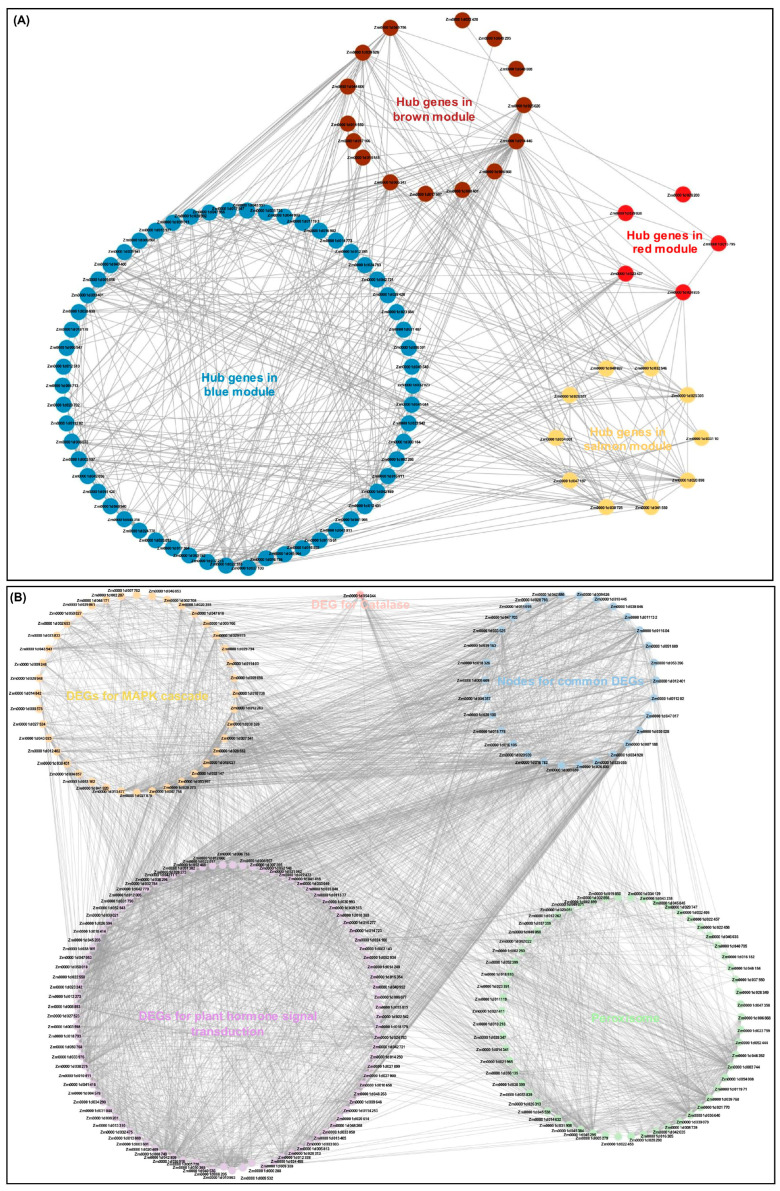
Genes network interaction construction. (**A**) Network interaction of hub genes in four modules (blue, brown, salmon, and red). (**B**) Network interaction of differentially expressed genes (DEGs) for peroxisome, catalase, plant hormone signal transduction, and MAPK cascade pathways. Nodes are the common DEGs involved in plant hormone signal transduction and MAPK cascade pathways.

**Figure 8 ijms-25-09396-f008:**
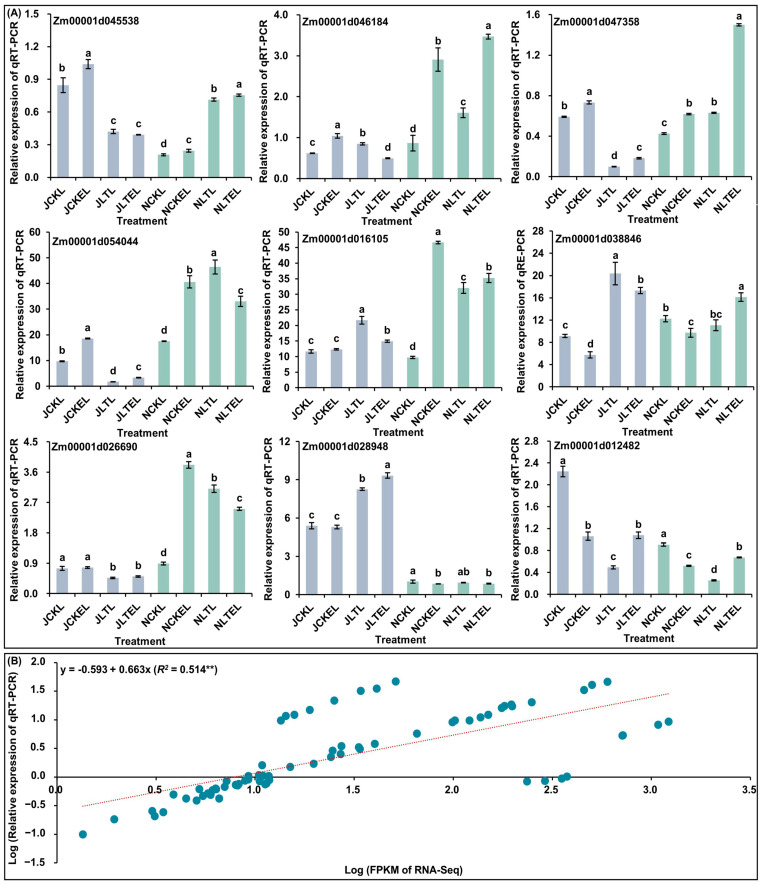
Quantitative real-time PCR (qRT-PCR) analysis of nine differentially expressed genes (DEGs) in leaves of both N192 and Ji853 seedlings at the three-leaf stage under four treatments. NCKL: leaves of N192 seedlings treated with 0 μM 2,4-epibrassinolide (EBR) application at 25 °C normal temperature (CK); NCKEL: leaves of N192 seedlings treated with 2 μM EBR application at 25 °C normal temperature (CKE); NLTL: leaves of N192 seedlings treated with 0 μM EBR application at 15 °C low-temperature (LT); NLTEL: leaves of N192 seedlings treated with 2 μM EBR application at 15 °C normal temperature (LTE); JCKL: leaves of Ji853 seedlings at CK; JCKEL: leaves of Ji853 seedlings at CKE; JLTL: leaves of Ji853 seedlings at LT; JLTEL: leaves of Ji853 seedlings at LTE. FPKM: the fragments per kilobase per million mapped. (**A**) The relative expression levels of nine DEGs in leaves of both maize genotypes under all treatments. Different lowercase letters in the same genotype under all treatments represent significant differences (*p* < 0.05) by analysis of variance (ANOVA). (**B**) Correlation between qRT-PCR and RNA-Seq for selected nine DEGs. ** represents significant difference (*p* < 0.01) by ANOVA.

**Figure 9 ijms-25-09396-f009:**
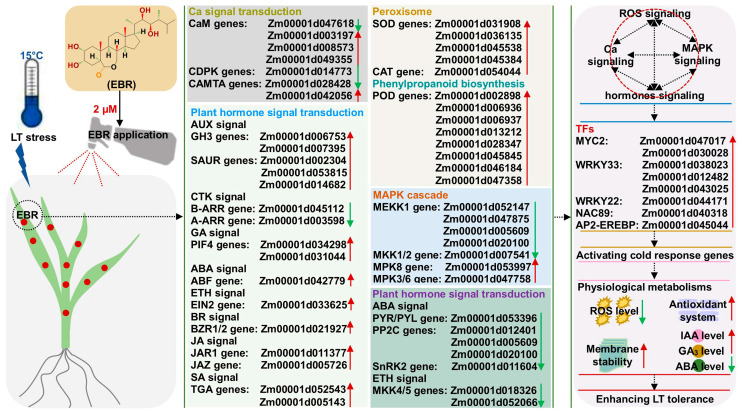
Schematic model of hypothetical mechanism for 2 μM 2,4-epibrassinolide (EBR) enhances low-temperature (LT) tolerance in maize seedlings. The red and green arrows represent up-regulated and down-regulated differentially expressed genes (DEGs)/transcription factors (TFs)/metabolites.

## Data Availability

Data are contained within the article and Appendix A.

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
