# Peer review of "Transcriptomic and Physiological Studies Unveil that Brassinolide Maintains the Balance of Maize’s Multiple Metabolisms under Low-Temperature Stress"

_ijms, 2024, doi:10.3390/ijms25179396_

Round 1
Reviewer 1 Report
Comments and Suggestions for Authors
The publication provides a comprehensive analysis of the physiological and molecular mechanisms by which 2,4-epibrassinolide enhances low-temperature tolerance in maize seedlings, using a combination of RNA sequencing, WGCNA, and physiological assays. This is a highly interesting study. I have a few recommendations for the authors:
- Abstract: It is recommended that the authors emphasize the key results more strongly, such as the DEGs involved in ROS homeostasis, plant hormone signaling, and the MAPK cascade.
- Introduction: The authors should specify the number of hub genes mentioned in line 74.
- Introduction: It is suggested that the authors provide the full names of MYB, NAC, and bZIP in line 78.
- Results: The full name of H2O2 should be provided when it first appears.
- Results: It is recommended to delete the phrase “The colored boxes represent corresponding DEGs” in lines 279-280 and 317.
- Discussion: The discussion is quite extensive; some content could be removed without compromising the quality of the publication.
Author Response
Dear Editor and Reviewers
Thank you for your letter of – and for the referee’s comments concerning our manuscript, “Transcriptomic and Physiological Studies Unveil Brassinolide Maintains Maize Multiple Metabolisms Balance under Low-Temperature Stress (Manuscript ID: ijms-3178405)”. We have carefully studied these comments and have made corresponding corrections to the manuscript, which we describe in detail below. We would like to re-submit the manuscript and that for possible publication on the Special Issue: “Plant Development and Hormonal Signaling” of International Journal of Molecular Sciences. Thank you very much for your time and consideration.
Editor:
Your manuscript has now been reviewed by experts in the field and can be found with the review reports at: https://susy.mdpi.com/user/manuscripts/resubmit/b0aa56b7bef0a99dd4fe8c1bf66a47d6 Please revise the manuscript found at the above link according to the reviewers' comments and upload the revised file within 5 days.
Thanks for the positive comments of you and all reviewers for our manuscript. As suggested, we have carefully revised and improved our manuscript using the “Track Changes” function of the manuscript at the above link. We then have re-submitted the manuscript within the allotted time.
Thank you for your consideration.
(I) Ensure all references are relevant to the content of the manuscript.
Thanks for the positive comments. As suggested, we have carefully checked all references. We then have re-submitted the manuscript.
Thank you for your consideration.
(II) Highlight any revisions to the manuscript, so editors and reviewers can see any changes made.
Thanks for the positive comments. As suggested, we have carefully revised and improved our manuscript using the “Track Changes” function of the manuscript. We then have re-submitted the manuscript.
Thank you for your consideration.
(III) Provide a cover letter to respond to the reviewers’ comments and explain, point by point, the details of the manuscript revisions.
Thanks for your positive comments for our manuscript. As suggested, we have carefully revised and improved our manuscript. In addition, we have prepared a detailed response letter to all reviewers for each point, and then have re-submitted the manuscript.
Thank you for your consideration.
(IV) If the reviewer(s) recommended references, critically analyze them to ensure that their inclusion would enhance your manuscript. If you believe these references are unnecessary, you should not include them.
Thanks for your positive comments for our manuscript. As suggested, we have carefully checked and revised the References. At the same time, we also have re-added two new references to enhance the quality of our manuscript. We then have re-submitted the manuscript.
Thank you for your consideration.
(V) If you found it impossible to address certain comments in the review reports, include an explanation in your appeal.
Thanks for your positive comments for our manuscript. As suggested, we have carefully revised and improved our manuscript. In addition, we have prepared a detailed response letter to all reviewers for each point, and then have re-submitted the manuscript.
Thank you for your consideration.
If your manuscript requires improvement to the language and/or figures, you may consider MDPI Author Services: https://www.mdpi.com/authors/english. Please note the status of this invitation “Publish Author Biography on the webpage of the paper” - https://susy.mdpi.com/user/manuscript/author_biography/b0aa56b7bef0a99dd4fe8c1bf66a47d6. If you wish to publish your biography, please complete it before your manuscript is accepted.
Thanks for the positive comments. As suggested, we have carefully checked and revised the English language of the manuscript. We then re-submitted the manuscript.
In addition, thanks for your invitation, we decided not to publish our biography.
Thank you for your consideration.
Please do not hesitate to contact us if you have any questions regarding the revision of your manuscript or if you need more time. We look forward to hearing from you soon.
Thanks for your positive comments for our manuscript. As suggested, we have carefully revised and improved the manuscript using the “Track Changes” function of our manuscript at the above link. We then have re-submitted the manuscript within the allotted time.
Thank you for your consideration.
Reviewer 1:
The publication provides a comprehensive analysis of the physiological and molecular mechanisms by which 2,4-epibrassinolide enhances low-temperature tolerance in maize seedlings, using a combination of RNA sequencing, WGCNA, and physiological assays. This is a highly interesting study. I have a few recommendations for the authors:
Thanks for your positive comments.
Thank you for your consideration.
- Abstract: It is recommended that the authors emphasize the key results more strongly, such as the DEGs involved in ROS homeostasis, plant hormone signaling, and the MAPK cascade.
Thanks for your positive comments. As suggested, we have provided the number of DEGs involved in ROS homeostasis, plant hormone signaling, and MAPK cascade, namely: “Transcriptome analysis revealed 332 differentially expressed genes (DEGs) were enriched in ROS homeostasis, plant hormone signal transduction, and mitogen-activated protein kinase (MAPK) cascade” in Lines 20-22 of Abstract section. We then have re-submitted the manuscript.
Thank you for your consideration.
- Introduction: The authors should specify the number of hub genes mentioned in line 74.
Thanks for your positive comments. As suggested, we have provided the number of hub genes in Introduction section, namely: “Liang et al. [24] found four hub genes involved in Ca2+ transport in maize in response to salt stress” in Lines 73-74 of Introduction section. We then have re-submitted the manuscript.
Thank you for your consideration.
- Introduction: It is suggested that the authors provide the full names of MYB, NAC, and bZIP in line 78.
Thanks for your positive comments. As suggested, we have provided the full names of MYB, NAC, and Bzip, namely: “with drought-related hub genes included myeloblastosis (MYB), NAM-ATAF1/2-CUC2 (NAC), and basic (region) leucine zippers (bZIP) transcription factors (TFs).” In Lines 78-79 of Introduction section. We then have re-submitted the manuscript.
Thank you for your consideration.
- Results: The full name of H2O2 should be provided when it first appears.
Thanks for your positive comments. As suggested, we have provided the full name of H2O2, namely: “Compared with 25°C normal temperature (CK), the rate changes (RCs) of O2•− production rate, O2•− content, and hydrogen peroxide (H2O2) content in N192/Ji853 leaves under LT stress ranged from 14.5% to 45.4% (Figure 1B).” in Lines 92-94 of Results section. We then have re-submitted the manuscript.
Thank you for your consideration.
- Results: It is recommended to delete the phrase “The colored boxes represent corresponding DEGs” in lines 279-280 and 317.
Thanks for your positive comments. As suggested, we have deleted the corresponding contents. We then have re-submitted the manuscript.
Thank you for your consideration.
- Discussion: The discussion is quite extensive; some content could be removed without compromising the quality of the publication.
Thanks for your positive comments. As suggested, we have revised and improved the corresponding content of Discussion section. We then have re-submitted the manuscript.
Thank you for your consideration.
Open Review: I would not like to sign my review report.
Thanks for your positive comments.
Thank you for your consideration.
Quality of English Language: I am not qualified to assess the quality of English in this paper.
Thanks for your positive comments.
Thank you for your consideration.
Does the introduction provide sufficient background and include all relevant references? Yes.
Thanks for your positive comments.
Thank you for your consideration.
Is the research design appropriate? Can be improved.
Thanks for your positive comments.
Thank you for your consideration.
Are the methods adequately described? Yes.
Thanks for your positive comments.
Thank you for your consideration.
Are the results clearly presented? Yes.
Thanks for your positive comments.
Thank you for your consideration.
Are the conclusions supported by the results? Yes.
Thanks for your positive comments.
Thank you for your consideration.
Reviewer 2:
This paper focuses on studies that reveal that brassinolide maintains the balance of multiple metabolisms in maize under stress conditions. The authors draw attention to abiotic stress induced by low temperature. The paper characterizes changes in the physiology and transcriptome of N192 and Ji853 seedlings at the three-leaf stage with or without 2μM 2,4-epibrassinolide (EBR) in environments of 25 and 15°C using high-performance liquid chromatography and RNA sequencing.
Thanks for your positive comments.
Thank you for your consideration.
The results are quite informative in terms of understanding the physiological and molecular mechanisms by which brassinolides enhance maize tolerance to low-temperature stress.
Thanks for your positive comments.
Thank you for your consideration.
The objectives of the paper are stated in the introductory part. The research methods are appropriate and clearly described. The structure of the paper (its composition and division into chapters) is correct. Tables and figures are adequate to the main text and readable. The abstract presents the aim, methods and results of the paper. The conclusion summarizes the main results and contributions to the current state of knowledge. The article includes relevant literature. The language and terms used are correct.
Thanks for your positive comments.
Thank you for your consideration.
However, there are a few suggestions to consider in advance:
Thanks for your positive comments. As suggested, we have revised and improved the manuscript. We then have re-submitted the manuscript.
Thank you for your consideration.
Introduction section: Please provide more detailed information about why this research is new.
Thanks for your positive comments. As suggested, we have provided the detailed information about why this research is new, namely:
Brassinolides (BRs), a class of an efficient, safe, and non-toxic hormones, are a collection of approximately 40 sterol compounds that are ubiquitous in plants [13]. They exhibits distinctive biological effects on normal plant growth and development and resistance to abiotic stresses, including seed germination, cell elongation, dark morphogenesis, flowering, senescence, quality formation, and stress resistance of salt, heavy metals, and drought [13–19]. Meanwhile, BRs have also attracted extensive attention in alleviating LT stress in recent years. 0.1 mg L-1 2,4-epibrassinolide (EBR) has been proven to reduce the cold damage symptoms of wheat (Triticum aestivum L.) by enhancing antioxidant enzymes activities, and decreasing malondialdehyde (MDA) formation at -10°C environment [20]. 0.3 mg L-1 EBR seed priming prevents rice (Oryza sativa L.) seedlings from chilling-induced oxidative stress by maintaining abscisic acid (ABA) and gibberellin 3 (GA3) levels at a 12/15°C day/night treatment [21]. Exposure to 10/5 °C day/night condition, 0.1 μM EBR improves photosynthesis of cucumber (Cucumis sativa L.) seedlings by promoting the activities of ribulose-1,5-biphosphate carboxylase/oxygenase, fructose-1,6-bisphosphate aldolase, sedoheptulose-1,7-bisphosphatase, and transketolase [22]. However, the effect BRs on maize seedlings growth and physiology, as well as the underlying molecular mechanisms under cold stress remains unclear.
Empowered by the development of high-throughput RNA-sequencing (RNA-Seq), the molecular mechanisms response to various stress pressures in maize are elucidated, and multiple crucial candidate genes were further identified at a large-scale. Additionally, weighted gene co-expression network analysis (WGCNA), a system biology method that characterizes the correlation pattern among genes, facilitates the understanding of gene associations and gene function and from RNA-Seq data [23]. So far this method has been widely employed in abiotic stress resistance analysis in maize, and several reliable hub genes were detected in the studies. For instance, using WGCNA, Liang et al. [24] found four hub genes involved in Ca2+ transport in maize in response to salt stress; Zhao et al. [25] determined 19 and 49 hub genes that regulated maize mesocotyl and coleoptile elongation plasticity under different light spectral quality stimulation. Cao et al. [26] identified two co-expression modules with significantly and highly correlated drought-related physiological indicators, with drought-related hub genes included myeloblastosis (MYB), NAM-ATAF1/2-CUC2 (NAC), and basic (region) leucine zippers (bZIP) transcription factors (TFs). At present, we know little about gene expression changes affected by EBR induced maize seedlings under LT stress and corresponding regulating hub genes.
We then have re-submitted the manuscript.
Thank you for your consideration.
References section: In the work, the authors used 66 sources. You can also cite researchers from Europe or America.
Thanks for your positive comments. As suggested, we have provided a new reference [67] from America. We then have re-submitted the manuscript.
Thank you for your consideration.
There are minor editorial errors in the work, e.g. Line: 769.
Thanks for your positive comments. As suggested, we have revised the corresponding content, namely: “Castroverde, C.D.M.; Dina, D. Temperature regulation of plant hormone signaling during stress and development. J. Exp. Bot. 2021, 72, 257.” We then have re-submitted the manuscript.
Thank you for your consideration.
Overall assessment: The work is very interesting, it will be interesting, not only for experts in the field. In my opinion, a wide range of researchers can benefit from it.
Thanks for your positive comments. As suggested, we have revised and improved the manuscript. We then have re-submitted the manuscript.
Thank you for your consideration.
I recommend its publication without hesitation after minor corrections.
Thanks for your positive comments.
Thank you for your consideration.
Open Review: I would like to sign my review report.
Thanks for your positive comments.
Thank you for your consideration.
Quality of English Language: I am not qualified to assess the quality of English in this paper.
Thanks for your positive comments.
Thank you for your consideration.
Does the introduction provide sufficient background and include all relevant references? Yes.
Thanks for your positive comments.
Thank you for your consideration.
Is the research design appropriate? Yes.
Thanks for your positive comments.
Thank you for your consideration.
Are the methods adequately described? Yes.
Thanks for your positive comments.
Thank you for your consideration.
Are the results clearly presented? Yes.
Thanks for your positive comments.
Thank you for your consideration.
Are the conclusions supported by the results? Yes.
Thanks for your positive comments.
Thank you for your consideration.
Sincerely,
Xiaoqiang Zhao professor
State Key Laboratory of Aridland Crop Science, Gansu Agricultural University
E-mail: zhaoxq3324@163.com

Reviewer 2 Report
Comments and Suggestions for Authors
This paper focuses on studies that reveal that brassinolide maintains the balance of multiple metabolisms in maize under stress conditions. The authors draw attention to abiotic stress induced by low temperature. The paper characterizes changes in the physiology and transcriptome of N192 and Ji853 seedlings at the three-leaf stage with or without 2μM 2,4-epibrassinolide (EBR) in environments of 25 and 15°C using high-performance liquid chromatography and RNA sequencing.
The results are quite informative in terms of understanding the physiological and molecular mechanisms by which brassinolides enhance maize tolerance to low-temperature stress.
The objectives of the paper are stated in the introductory part. The research methods are appropriate and clearly described. The structure of the paper (its composition and division into chapters) is correct. Tables and figures are adequate to the main text and readable. The abstract presents the aim, methods and results of the paper. The conclusion summarizes the main results and contributions to the current state of knowledge. The article includes relevant literature. The language and terms used are correct.
However, there are a few suggestions to consider in advance:
Introduction section: Please provide more detailed information about why this research is new.
References section: In the work, the authors used 66 sources. You can also cite researchers from Europe or America.
There are minor editorial errors in the work, e.g. Line: 769.
Overall assessment: The work is very interesting, it will be interesting, not only for experts in the field. In my opinion, a wide range of researchers can benefit from it.
I recommend its publication without hesitation after minor corrections.
Author Response

(The authors gave the same response as above.)
